# Microglia Depletion Attenuates the Pro-Resolving Activity of the Formyl Peptide Receptor 2 Agonist AMS21 Related to Inhibition of Inflammasome NLRP3 Signalling Pathway: A Study of Organotypic Hippocampal Cultures

**DOI:** 10.3390/cells12212570

**Published:** 2023-11-03

**Authors:** Kinga Tylek, Ewa Trojan, Monika Leśkiewicz, Imane Ghafir El Idrissi, Enza Lacivita, Marcello Leopoldo, Agnieszka Basta-Kaim

**Affiliations:** 1Laboratory of Immunoendocrinology, Department of Experimental Neuroendocrinology, Maj Institute of Pharmacology, Polish Academy of Sciences, 12 Smętna St., 31-343 Kraków, Poland; tylek@if-pan.krakow.pl (K.T.); trojan@if-pan.krakow.pl (E.T.); leskiew@if-pan.krakow.pl (M.L.); 2Department of Pharmacy—Drug Sciences, University of Bari, Via Orabona 4, 70125 Bari, Italy; imane.ghafir@uniba.it (I.G.E.I.); enza.lacivita@uniba.it (E.L.); marcello.leopoldo@uniba.it (M.L.)

**Keywords:** formyl peptide receptor 2, ureidopropanamide agonist, neuroinflammation, lipopolysaccharide, inflammasome NLPR3-related pathway, hippocampus

## Abstract

Microglial cells have been demonstrated to be significant resident immune cells that maintain homeostasis under physiological conditions. However, prolonged or excessive microglial activation leads to disturbances in the resolution of inflammation (RoI). Formyl peptide receptor 2 (FPR2) is a crucial player in the RoI, interacting with various ligands to induce distinct conformational changes and, consequently, diverse biological effects. Due to the poor pharmacokinetic properties of endogenous FPR2 ligands, the aim of our study was to evaluate the pro-resolving effects of a new ureidopropanamide agonist, compound AMS21, in hippocampal organotypic cultures (OHCs) stimulated with lipopolysaccharide (LPS). Moreover, to assess whether AMS21 exerts its action via FPR2 specifically located on microglial cells, we conducted a set of experiments in OHCs depleted of microglial cells using clodronate. We demonstrated that the protective and anti-inflammatory activity of AMS21 manifested as decreased levels of lactate dehydrogenase (LDH), nitric oxide (NO), and proinflammatory cytokines IL-1β and IL-6 release evoked by LPS in OHCs. Moreover, in LPS-stimulated OHCs, AMS21 treatment downregulated NLRP3 inflammasome-related factors (CASP1, NLRP3, PYCARD) and this effect was mediated through FPR2 because it was blocked by the FPR2 antagonist WRW4 pre-treatment. Importantly this beneficial effect of AMS21 was only observed in the presence of microglial FPR2, and absent in OHCs depleted with microglial cells using clodronate. Our results strongly suggest that the compound AMS21 exerts, at nanomolar doses, protective and anti-inflammatory properties and an FPR2 receptor located specifically on microglial cells mediates the anti-inflammatory response of AMS21. Therefore, microglial FPR2 represents a promising target for the enhancement of RoI.

## 1. Introduction

Microglia, the brain’s resident immune cells, play a crucial role in health and disease [1,2]. They defend the brain against infections and injury [3], contribute to neurogenesis and synaptic conditions, and interact with neurons and other brain cells to control homeostasis [4,5,6,7]. Importantly, microglial cells are susceptible to the surrounding environment [8] and possess various receptors to sense microenvironment changes that regulate their functions [9,10]. Therefore, during inflammatory stimulation, the brain environment plays a crucial role in the proper resolution of inflammation (RoI). In fact, triggering microglial reactivity and prolonged stimulation, they alter homeostasis, leading to neuronal and other glial tissue damage [11,12,13]. Once activated, microglia change their morphology, proliferate and enhance the expression of various cluster of differentiation (CD) markers (e.g., CD40 and CD68). Moreover, microglia are a prominent source of proinflammatory cytokines such as IL-1β, IL-18, IL-6 and tumour necrosis factor alpha (TNF-α), as well as neurotoxic mediators including NO, ROS, prostaglandin (PG) E2, and the superoxide anion [14,15].

One of the crucial players in the development of the neuroinflammatory response is the nucleotide-binding oligomerisation domain-like (NOD-like) receptor pyrin-containing 3 inflammasome (NLRP3), which is highly expressed in microglia [16,17,18]. NLRP3 is a multiprotein complex consisting of NLRP3, procaspase-1, and apoptosis-associated speck-like protein containing a caspase recruitment domain (PYCARD). Various environmental and endogenous molecules activate the NLRP3 complex. Additionally, this complex is indirectly activated by a primary component of the endotoxin from Gram-negative bacterial cell walls, lipopolysaccharide (LPS) [19,20]. The two-step NLRP3 inflammasome activation facilitates the oligomerisation of inactive NLRP3, PYCARD, and procaspase-1. This complex, in turn, catalyses the conversion of procaspase-1 to caspase-1, which contributes to the production and secretion of microglial mature proinflammatory cytokines, mainly IL-1β and IL-18 [21,22,23].

In addition, to proinflammatory polarisation, microglial cells also have an anti-inflammatory phenotype and are involved in the suppression of inflammation and in the restoration of homeostasis inter alia by the release of transforming growth factor (TGF-β), insulin-like growth factor 1 (IGF-1), and IL-10 [24,25,26]. Alterations in microglial polarisation are associated with excessive microglial inflammatory activation and disturbances in the RoI [11,13]. Since microglial cells are highly heterogeneous, as identified via novel transcriptomic and microscopy studies [27,28], it has been suggested that by targeting specific receptors, microglial cells can be “reprogrammed” to adopt new functional and molecular identities in a context-dependent manner [29].

Formyl peptide receptors (FPRs) belong to the seven-transmembrane G protein-coupled receptor family. As demonstrated in previous scientific research in the brain, FPR2 is expressed mainly in microglial cells, neurons, and astroglia not only in physiological but also pathological conditions [30,31,32,33,34,35]. Among them, FPR2 is a low-affinity receptor for N-formyl peptides and is considered the most promiscuous member of the FPR family. FPR2 can recognise various endogenous and exogenous ligands, ranging from lipids to proteins and peptides, including non-formylated peptides [36,37]. FPR2 also shows diversity in receptor signalling leading to different and sometimes opposite downstream effects, which have been ascribed to conformational changes that underline biased signalling [38,39]. Biased agonism explains at least in part the role of different FPR2 agonists in modulating the inflammatory response [40]. In fact, after the binding of the ligand, FPR2 is activated and triggers several agonist-dependent signal transduction pathways [41,42]. However, it should be strongly emphasised that FPR2 downstream signalling pathway activation depends not only on the ligand but also on the cell type involved [43,44], which is crucial in understanding how FPR2 activation can modulate cellular processes, including the RoI.

Therefore, identifying new bioactive compounds that can target microglial FPR2 and inflammatory pathways to dampen the neuroinflammatory response may be a useful approach in preventing or delaying the onset of neuroinflammation-based brain diseases.

We recently identified a series of ureidopropanamide-based FPR2 agonists endowed with high agonist potency. Among the studied compounds, derivative AMS21 ((S)-1-(3-(4-cyanophenyl)-1-(indolin-1-yl)-1-oxopropan-2-yl)-3-(4-fluorophenyl)urea) was able to activate FPR2 at nanomolar concentrations (EC50 = 0.026 μM), which was was 10-fold selective over FPR1 subtype and showed acceptable resistance to oxidative metabolism, which are promising properties for prospective in vivo preclinical studies [45]. In addition, in rat primary microglial cells stimulated with LPS, the anti-inflammatory effects of AMS21 were counterbalanced by the FPR2 antagonist WRW4, suggesting these were mainly mediated through the interaction with FPR2 [45].

In the present study, we assessed the pro-resolving effects of compound AMS21 on the neuroinflammatory response stimulated by bacterial endotoxin treatment using organotypic hippocampal cultures (OHCs). Moreover, to confirm the contribution of FPR2 activation to the observed effects of AMS21, we used the FPR2 receptor antagonist WRW4. Next, we focused on the impact of AMS21 on FPR2 localised on microglial cells, assessing the anti-inflammatory activity of this ligand in OHCs depleted of microglial cells using clodronate. The intracellular mechanisms of the effects of AMS21, with particular emphasis on the pathway associated with the NLRP3 inflammasome under standard conditions and in microglia-deficient cultures, were also evaluated.

## 2. Materials and Methods

### 2.1. Animals

Sprague Dawley rats were obtained from Charles Rivers (Sulzfel, Germany) and kept under standard conditions. The rats were maintained at room temperature (23 °C) on a 12/12 h light/dark cycle (lights on at 06:00 am) with food and water available ad libitum. After one week of acclimatisation, the presence of the proestrus cycle was identified by taking vaginal smears daily. On the proestrus day, females and males were mated overnight. Next, pregnant females were kept under standard conditions in home cages without any disruptions. The experiments were approved by the Committee for Laboratory Animal Welfare and Ethics of the Maj Institute of Pharmacology, Polish Academy of Sciences, Cracow, Poland (approval no. 204/2018, 28 June 2018).

### 2.2. Chemicals

The formyl peptide receptor 2 agonist compound AMS21 ((S)-1-(3-(4-cyanophenyl)-1-(indolin-1-yl)-1-oxopropan-2-yl)-3-(4-fluorophenyl)urea) was synthesised at the Department of Pharmacy, University of Bari, as previously reported [45]. The FPR2 antagonist WRW4 was purchased from Alomone Labs, Israel. The bacterial endotoxin lipopolysaccharide (LPS; *Escherichia coli* 0111:B4) was obtained from Sigma-Aldrich, St. Louis, MO, USA. Microglial cells were depleted using clodronate (disodium salt) obtained from Millipore, Burlington, MA, USA. Stock solutions of the used compounds were prepared as follows: AMS21 in DMSO to 1 mM concentration, LPS in PBS to 1 mg/mL, WRW4, and clodronate in distilled water to 1 mM and 1 mg/mL concentration, respectively. The final concentrations of the used compounds were in distilled water.

### 2.3. Establishment of Organotypic Hippocampal Cultures (OHCs)

Organotypic hippocampal cultures were prepared according to the protocol of Stoppini et al. [46] with slight modifications. The cultures were prepared from six- to seven-day-old Sprague Dawley females. The animals were decapitated, and the isolated brains were placed directly into a sterile ice-cold working buffer (96% HBSS, 3.5% glucose, and 0.5% penicillin/streptomycin; all reagents were obtained from Gibco, Waltham, MA, USA). Afterwards, the hippocampi were placed on Teflon discs and cut into 350 μm sections using a McIlwain™ Tissue Chopper (Surrey, UK). Then, selected sections were transferred into ThinCerts™ (Minneapolis, MN, USA) inserts with 0.4 μm pore size membranes (Greiner Bio-one, Kremsmunster, Austria) in 6-well plates containing 1 mL of initial medium with 25% horse serum (50% DMEM + GlutaMax™-I, pH 7.4; 20.5% HBSS; 25% horse serum; 0.1 mg/mL glucose; 1% amphotericin B; 0.4% penicillin and streptomycin; 1% B-27 supplement; and HEPES (all reagents were obtained from Gibco, London, UK)). OHCs were grown for 7 days in vitro (DIV) in an incubator (37 °C) with an adjustable CO_2_ flow (5%). The culture medium was changed every 48 h, and horse serum concentration was tapered down according to the following schedule: on 1st to 3rd DIV 25% medium; on 5th DIV 10% medium (50% DMEM + GlutaMax™-I, pH 7.4; 35.5% HBSS; 10% horse serum; 0.1 mg/mL glucose; 1% amphotericin B; 0.4% penicillin and streptomycin; 1% B-27 supplement; and HEPES, (all reagents were obtained from Gibco, London, UK)); and lastly on 7th DIV serum-free medium (50% DMEM F-12, pH 7.4; 44% HBSS; 0.1 mg/mL glucose; 1% amphotericin B; 0.4% penicillin and streptomycin, 1% B-27, 1% N-2; and HEPES).

### 2.4. Treatment

Treatment of OHCs was divided into two parts depending on clodronate presence or absence. In the groups without clodronate on the 7th DIV, pre-treatment was performed with the FPR2 antagonist WRW4 (10 μM) for 30 min. Then, the new FPR2 agonist compound AMS21 at two doses, 0.1 μM and 1 μM, was added to the culture medium for an hour, and, afterwards, OHCs were stimulated with 1 μg/mL LPS for 24 h.

In the second part of the experiments, to deplete microglia, clodronate was prepared and used according to the manufacturer’s instructions. Clodronate was administered to the culture medium in the 1st DIV for 24 h. Afterwards, the culture medium was removed, 6-well plates were washed twice with warm, sterile PBS, and a new medium was added. On the 7th DIV, OHCs were treated with AMS21 at a dose of 0.1 μM (selected in the first part of the experiments), and an hour later, bacterial endotoxin at 1 μg/mL dose was added for 24 h. Control OHCs were treated with PBS as a vehicle.

### 2.5. Lactate Dehydrogenase (LDH) Assay

A lactate dehydrogenase (LDH) assay was conducted using a Cytotoxicity Detection Kit (Roche, Germany) as previously described [47,48]. Briefly, 24 h after LPS administration, the culture medium was collected, and 50 μL of each sample was placed into a 96-well plate. Then, an equal amount of reagent mixture prepared according to the manufacturer’s instructions was mixed with the samples. After incubation at 37 °C, the intensity of the red colour formed in the colorimetric assay was measured at a wavelength of 490 nm (Infinite^®^ 200 PRO plate reader, Tecan, Zurich, Switzerland) and was proportional to the number of damaged/dead cells.

### 2.6. Nitric Oxide (NO) Assay

The amount of nitric oxide (NO) was detected as we previously described [49,50] using a colorimetric Griess reaction in accordance with the protocol. An equal volume of the collected samples (50 μL), Griess A (0,1% N-1-naphthylethylenediamine dihydrochloride), and Griess B (1% sulfanilamide in 5% phosphoric acid; Sigma-Aldrich, St. Louis, MO, USA), was mixed in a 96-well plate. The intensity of the formed colour was measured at a wavelength of 540 nm (Infinite^®^ 200 PRO plate reader, Tecan, Zurich, Switzerland).

### 2.7. RNA Extraction and cDNA Preparation

Twenty-four hours after LPS administration, slices were lysed using 200 μL TRI^®^ Reagent (Sigma-Aldrich, St. Louis, MO, USA) and stored at −20 °C until isolation. Total RNA extraction was performed according to the User Guide (TRIzol^®^ Reagent User Guide Instructions; Thermo Fisher Scientific, Waltham, MA, USA) based on the Chomczynski [51] method. Then, the RNA concentration was assessed using a NanoDrop spectrophotometer (ND/1000 UV/Vis, Thermo Fisher NanoDrop, Waltham, MA, USA). The synthesis of cDNA was performed using an NG dART RT Kit (EURx, Gdansk, Poland) according to the manufacturer’s instructions.

### 2.8. Quantitative Real-Time Polymerase Chain Reaction (qRT-PCR)

The cDNA was amplified using a FastStart Universal Probe Master (Rox) kit (Roche, Basel, Switzerland) and TaqMan probes (Thermo Fisher Scientific, Waltham, MA, USA) for the following genes: *Casp1* (*Caspase 1*; Rn00562724_m1), *Cd40* (*Cluster of differentiation 40*; Rn01423590_m1), *Cd68* (*Cluster of differentiation 68*; Rn01495634_g1), *Igf-1* (*Insulin-like growth factor 1*; Rn00710306_m1), *Il-1ra* (*Interleukin 1 receptor antagonist*; Rn02586400_m1), *Il-1β* (*Interleukin 1β*; Rn00580432_m1), *Il-6* (*Interleukin 6*; Rn01410330_m1), *Il-18* (*Interleukin 18*; Rn01422083_m1), *Nlrp3* (*NLR family pyrin domain containing 3*; Rn04244620_m1), *Pycard* (*Apoptosis-associated speck-like protein containing a caspase recruitment domain*; Rn00597229_g1), *Tgf-β1* (*Transforming growth factor β1*; Rn00572010_m1) (all obtained from Thermo Fisher Scientific, Waltham, MA, USA). The *B2m* gene (*β2 microglobulin*; Rn00560865_m1) was used as a normalising control. Thermal cycling conditions were as follows: 2 min at 50 °C and 10 min at 95 °C, followed by 40 cycles at 95 °C for 15 s and at 60 °C for 1 min. The threshold value (Ct) for each sample was set in the exponential phase of PCR, and the ∆∆Ct method was used for data analysis.

### 2.9. Enzyme-Linked Immunosorbent Assay (ELISA)

The OHCs medium was collected 24 h after LPS administration for further experiments. Furthermore, OHCs were lysed using 160 μL of RIPA buffer with Halt™ Protease and Phosphatase Inhibitor Cocktail (Thermo Fisher Scientific, Waltham, MA, USA). Protein isolation was performed, and the total concentration of the protein was assessed using a Pierce™ BCA Protein Assay Kit (Thermo Fisher Scientific, Waltham, MA, USA). Optical density was measured at a wavelength of 562 nm using an Infinite^®^ 200 PRO plate reader (Tecan, Zurich, Switzerland). The levels of interleukin 1β, interleukin 6, interleukin 10 and transforming growth factor β were measured in the collected supernatants, and the levels of caspase 1, NLR family pyrin domain containing 3, and apoptosis-associated speck-like protein containing a caspase recruitment domain were assessed in the protein isolated from OHCs. All tests were performed using commercially available enzyme-linked immunosorbent assays (ELISA) obtained from Wuhan Fine Biotech Co., Ltd. Wuhan, China (IL-1β, TGF-β, CASP1, NLRP3, PYCARD), BD Biosciences, Franklin Lakes, NJ, USA (IL-6, IL-10) in accordance with the manufacturer’s protocols. The detection limits were as follows: IL-1β < 18.75 pg/mL, IL-6 < 78 pg/mL, IL-10 < 15.6 pg/mL, TGF-β < 18.75 pg/mL, CASP1 < 37.5 pg/mL, NLRP3 < 0.188 ng/mL, and PYCARD < 46.875 pg/mL. The inter-assay precision of all ELISA kits was CV% < 10%. The intra-assay precision of all ELISA kits was CV% < 8%.

### 2.10. Western Blot Analyses

Western blot analyses were carried out as we described previously [4,6,52]. The samples (equal protein concentration) were mixed with Laemmli buffer (Roche, Basel, Switzerland) in a 4:1 ratio and heated in an Eppendorf Thermomixer comfort (Sigma-Aldrich, St. Louis, MO, USA) for 8 min at 95 °C. Afterwards, the samples were resolved by SDS-PAGE on 4–20% Criterion™ TGX™ Precast Gels (Bio-Rad, Hercules, CA, USA) and transferred to PVDF membranes (Sigma-Aldrich, St. Louis, MO, USA) using Trans-Blot Turbo (Bio-Rad, Hercules, CA, USA). Membranes with transferred protein were briefly washed with Tris-buffered saline (TBS) (pH 7.5), blocked in 5% bovine serum albumin (5% BSA dissolved in TBS with 0.1% Tween 20 (TBST)) for 1 h at room temperature (RT), and washed with TBST 3 times for 10 min. Then, the membranes were incubated with primary antibodies overnight at 4 °C anti-IBA1 (NBP2-19019, 1:500, Novus Biologicals, Centennial, CO, USA) and anti-vinculin (1:15,000, V9264, Sigma-Aldrich, St. Louis, MO, USA) diluted in Signal Boost Immunoreaction Enhancer Kit Buffer (Millipore, Warsaw, Poland). The next day, the membranes were washed in TBST 3 times for 10 min and incubated with horse anti-mouse IgG (1:10,000, PI-2000 Vector Laboratories, Newark, CA, USA) and goat anti-rabbit IgG (1:10,000, PI-1000, Vector Laboratories, Newark, CA, USA) secondary antibodies for an hour. Finally, the membranes were washed again with TBST, and blots were detected using Pierce™ ECL Western blotting substrate (Thermo Fisher, Waltham, MA, USA) and visualised using a Fujifilm LAS1000 system (Fuji Film, Tokyo, Japan). The relative levels of immunoreactivity were quantified using Fujifilm Multi Gauge V3.0 software (Fuji Film, Tokyo, Japan).

### 2.11. Immunofluorescence Staining of Organotypic Hippocampal Cultures

Immunofluorescence staining of OHCs was performed in accordance with Gogolla et al. [53] with slight modifications. Twenty-four hours after LPS administration, OHCs were fixed with 4% paraformaldehyde (PFA, ChemCruz^®^, Santa Cruz Biotechnology, Inc., Dallas, TX, USA) 1 mL above and 1 mL underneath the insert for an hour. The slices were washed with PBS 3 times for 5 min, and the sections were kept at 4 °C until experiments. Afterwards, OHCs were carefully removed from the inserts, placed into 12-well plates, and permeabilised in 0.5% Triton X-100 in PBS for up to 18 h at 4 °C. Then, the sections were blocked with 20% BSA in PBS solution (20% bovine serum albumin, Sigma-Aldrich, St. Louis, MO, USA) overnight at 4 °C or for 4 h at RT. All blocked sections were first incubated with an anti-FPR2 rabbit polyclonal antibody (Huabio, Greater Boston, MA, USA; 1:50) and then with a goat anti-rabbit antibody conjugated with the fluorescent dye AlexaFluor^®^ 647 (Abcam, Cambridge, UK; 1:300) overnight at 4 °C in a closed wet chamber. Secondary staining was carried out using the primary antibody anti-IBA1 goat polyclonal antibody (Abcam, Cambridge, UK; 1:50) under the same conditions. Subsequently, the slices were incubated with donkey anti-goat secondary antibody conjugated with the fluorescent dye Alexa Fluor^®^ 555 (Abcam, Cambridge, UK; 1:300). OHCs were briefly washed with 5% BSA in PBS and stained with Hoechst 33342 (Invitrogen, Waltham, MA, USA; 1:5000) for 15 min at RT to stain the nuclei. Dyed OHCs were placed onto microscope slides and mounted using ProLong™ Glass Antifade Mountant (Invitrogen, Waltham, MA, USA), covered with cover glass, and kept at 4 °C until imaging with a Leica TCS SP8 X confocal laser-scanning microscope (Leica Microsystems CMS GmbH, Mannheim, Germany) using a 63× HC PL APO CS2 1.40 NA oil immersion objective. The images were reconstructed using ImageJ 1.53n (Wayne Rasband, National Institute of Health, Bethesda, MD, USA).

### 2.12. Quantitative Analyses of Confocal Fluorescence Images of Organotypic Hippocampal Cultures

The fluorescence intensity of IBA1 and FPR2 was measured using ImageJ. Briefly, the intensity in single images was measured after applying the colour threshold separately for IBA1 and FPR2. Then, the thresholded area derived from the entire picture was determined, and the data are presented as the control mean ± SEM.

### 2.13. Statistical Analysis

The data were analysed using Statistica 13.3 software (Stat Soft, Tulsa, OK, USA). All biochemical experiments were conducted under the same conditions for all samples, regardless of the type of treatment. The results were obtained from independent experiments carried out under the same conditions and are presented as the mean ± SEM. Data obtained from LDH, NO, and confocal microscopy are presented as the mean percentage ± SEM of the control. The results obtained from the ELISA are presented as the mean ± SEM, and those for qRT-PCR are presented as the average fold change ± SEM. All groups were compared using factorial analysis of variance (ANOVA) to determine the effects of the factors, followed, when appropriate, by Duncan’s post hoc test. A *p* value of less than 0.05 was considered to be statistically significant. All graphs were prepared using GraphPad Prism 9.

## 3. Results

### 3.1. The Effect of WRW4 and AMS21 Treatment on Lactate Dehydrogenase and Nitric Oxide Release in OHCs

To establish the most efficient dose of the FPR2 agonist AMS21, our initial studies focused on the assessment of the release of lactate dehydrogenase (LDH), which is a cell death marker, and nitric oxide (NO) induced by LPS stimulation. For this purpose, we treated cultures with AMS21 at two doses: 0.1 μM and 1 μM. Furthermore, OHCs were also pre-treated with the FPR2 antagonist WRW4 to determine whether the observed effect of the tested agonist was mediated through FPR2.

Treatment of OHCs with AMS21 at both doses and WRW4 did not affect LDH release in the control groups (Figure 1A). However, we observed a decrease in NO release after combined AMS21 and WRW4 administration (*p* = 0.045149 and *p* = 0.034001 for 0.1 μM and 1 μM AMS21 doses, respectively) (Figure 1B). Stimulation of OHCs with LPS (1 μg/mL) elevated the levels of both LDH (*p* = 0.008050) (Figure 1A) and NO (*p* = 0.000033) (Figure 1B) release. LDH release was decreased after AMS21 administration in LPS-stimulated OHCs; however, this beneficial effect was observed only after a lower (0.1 μM) dose (*p* = 0.003820) (Figure 1A). Moreover, we observed a significant attenuation of NO release at 0.1 μM (*p* = 0.000057) and 1 μM (*p* = 0.000057) AMS21 doses (Figure 1B) in LPS-challenged groups. Pre-treatment with the FPR2 antagonist WRW4 did not significantly modulate favourable AMS21 properties in LPS-stimulated groups. Since AMS21 attenuated the release of LDH and NO only at 0.1 μM, this dose was chosen for further studies.

### 3.2. The Effect of WRW4 and AMS21 Treatment on the Release of the Proinflammatory Cytokines IL-1β and IL-6 in OHCs

To examine whether AMS21 has anti-inflammatory properties, we assessed the levels of two proinflammatory cytokines, IL-1β and IL-6, using ELISA. Our findings indicate that AMS21 (0.1 μM) and WRW4 (10 μM) did not affect the levels of IL-1β (Figure 2A) and IL-6 (Figure 2B) under basal conditions. As expected, LPS (1 μg/mL) stimulation significantly elevated the levels of both the proinflammatory cytokines IL-1β (*p* = 0.007455) (Figure 2A) and IL-6 (*p* = 0.000039) (Figure 2B). Furthermore, AMS21 attenuated the inflammatory response caused by LPS since decreased levels of IL-1β (*p* = 0.022065) (Figure 2A) and IL-6 (*p* = 0.029148) (Figure 2B) were observed. In the case of IL-1β, the anti-inflammatory effect of AMS21 was mediated via FPR2, and pre-treatment with WRW4 blocked the beneficial effect of AMS21 (*p* = 0.000732) (Figure 2A).

### 3.3. The Effect of WRW4 and AMS21 Treatment on the Release of Anti-Inflammatory Cytokines TGF-β and IL-10 in OHCs

In the next part of our research, we determined the pro-resolving properties of AMS21 by assessing the protein levels of two anti-inflammatory cytokines, TGF-β and IL-10. Treatment of OHCs with WRW4 (10 μM) and AMS21 (0.1 μM) under basal conditions reduced the protein level of TGF-β (*p* = 0.016505) (Figure 3B). Nevertheless, an equivalent effect was not observed in the case of IL-10 (Figure 3A) under basal conditions. Our findings indicate elevated levels of both cytokines TGF-β (*p* = 0.010386) (Figure 3A) and IL-10 (*p* = 0.021561) (Figure 3B) in LPS-stimulated groups. Moreover, AMS21 administration slightly increased the TGF-β (*p* = 0.000818) (Figure 3A) level and maintained an enhanced level of IL-10 (*p* = 0.018393) (Figure 3B) in the LPS-challenged group in comparison to the control group. Importantly, the favourable effect of AMS21 was mediated via FPR2 in the case of TGF-β release (*p* = 0.000025) (Figure 3A) since we observed a diminished level of this factor in WRW4 pre-treated groups.

### 3.4. The Effect of AMS21 on the mRNA Expression of Proinflammatory and Anti-Inflammatory Genes

Considering the favourable role of AMS21 in modulating anti-inflammatory and pro-resolving action and the fact that microglia are crucial immune cells in the central nervous system, we assessed proinflammatory and anti-inflammatory gene profiles with particular consideration of microglial markers.

We determined the mRNA expression levels of proinflammatory (*Cd40*, *Cd68*, *Il-1β*, *Il-6*, *Il-18*) (Table 1 part A) and anti-inflammatory genes (*Igf-1*, *Il-1Ra*, *Tgf-β*) (Table 1 part B). The statistical analysis of proinflammatory genes revealed the upregulation of the *Cd40* (*p* = 0.037870), *Il-1β* (*p* = 0.000159), *Il-6* (*p* = 0.036641), and *Il-18* (*p* = 0.028277) (Table 1 part A) genes in LPS-treated (1 μg/mL) OHCs. Importantly, the impact of AMS21 (0.1 μM) was noticed as the downregulation of *Cd40* (*p* = 0.042920 and *Il-18* (*p* = 0.030416) genes after LPS stimulation. As shown in Table 1 part B, LPS administration decreased *Igf-1* (*p* = 0.002364) and increased the *Il-1Ra* (*p* = 0.008516) mRNA levels. AMS21 potentiated the elevation of *Il-Ra* (*p* = 0.031202) evoked by LPS.

### 3.5. The Effect of WRW4 and AMS21 Treatment on the NLRP3-Related Pathway in OHCs

We also determined the effect of AMS21 on the protein level and mRNA expression of all NLRP3 inflammasome subunits, which is a multiprotein complex containing sensor protein (NLRP3), adaptor protein (PYCARD) and caspase-1 protein [21,54].

As revealed in Figure 4, AMS21 (0.1 μM) and WRW4 (10 μM) did not affect the protein level of all examined subunits under basal conditions. However, LPS (1 μg/mL) stimulation led to a significant increase in CASP1 (*p* = 0.000057) (Figure 4A), NLRP3 (*p* = 0.000028) (Figure 4B) and PYCARD (*p* = 0.000814) (Figure 4C) in OHCs. Moreover, AMS21 attenuated LPS-induced elevations in CASP1 (*p* = 0.000028), NLRP3 (*p* = 0.000055) and PYCARD (*p* = 0.018303), while pre-treatment with WRW4 blocked these beneficial properties of the agonist (CASP1 (*p* = 0.037574), NLRP3 (*p* = 0.000062), PYCARD (*p* = 0.006172)).

Analysis of mRNA expression revealed an elevated level of *Nlrp3* (*p* = 0.017181) and downregulation of *Pycard* (*p* = 0.044698) genes in LPS-stimulated OHCs (Figure 4D). Furthermore, we observed restored homeostasis in the *Nlrp3* (*p* = 0.011565) gene after AMS21 treatment in LPS-challenged groups. Although we did not observe statistical significance, *Casp1* tended to be elevated after LPS administration and restored after AMS21 treatment.

### 3.6. The Effect of Clodronate Treatment on Microglia in OHCs

Formyl peptide receptor 2 is widespread in peripheral immune cells and in central nervous system cells, including neurons, astrocytes, and microglia [30,31,32,33]. Since our research has shown that AMS21 exerts its biological function via FPR2, we determined whether the presence of microglial FPR2 is required for the anti-inflammatory and pro-resolving action of AMS21. As a preliminary part of these experiments, we confirmed that clodronate depleted microglia using immunofluorescence staining (in Appendix A Appendix A) and Western blot analysis.

As we demonstrated in Figure 5A, the fluorescence intensity of IBA1 in LPS-treated OHCs (1 μg/mL) was significantly increased (*p* = 0.005184). Moreover, after clodronate (150 μg/mL) administration, the fluorescence intensity decreased in both the control (*p* = 0.000443) and LPS-challenged group (*p* = 0.000061). This finding seems to be in line with the fluorescence intensity of FPR2 (Figure 5B), as we observed a diminished level of FPR2 in control (*p* = 0.0004026) and LPS-stimulated OHCs (*p* = 0.012620) after clodronate treatment. Nevertheless, the fluorescence intensity of FPR2 remains not completely silenced, as in the microglial marker IBA1.

In the second part of this experiment, to ensure that clodronate depleted microglia, we performed Western blot analysis (Figure 5C and Appendix A Appendix A). As expected, LPS administration elevated the protein level of IBA1 (*p* = 0.002265), and clodronate stimulation diminished the microglial marker in both control (*p* = 0.002506) and LPS-treated (*p* = 0.000070) OHCs.

### 3.7. The Effect of AMS21 on the Protein Level of NLRP3 Inflammasome Pathway-Related Factors in Microglia-Depleted OHCs

Finally, we investigated the proinflammatory and pro-resolving properties of AMS21 in microglia-depleted OHCs. Our research revealed that clodronate (150 μg/mL) and AMS21 (0.1 μM) administration did not change the protein levels of IL-1β (Figure 6A) and CASP1 (Figure 6B) under basal conditions. Nevertheless, we observed the influence of combined treatment with clodronate and AMS21 in vehicle-treated groups since the protein levels of NLRP3 (*p* = 0.015504) (Figure 6C) and PYCARD (*p* = 0.032702) (Figure 6D) were decreased compared to those in the AMS21-treated group. As expected, all examined factor levels were elevated in LPS-stimulated OHCs (IL-1β (*p* = 0.001421), CASP1 (*p* = 0.000054), NLRP3 (*p* = 0.000055), PYCARD (*p* = 0.000054)), and AMS21 exhibited favourable properties by decreasing the amount of NLRP3 inflammasome pathway-related factors (IL-1β (*p* = 0.016751), CASP1 (*p* = 0.000028), NLRP3 (*p* = 0.000032), PYCARD (*p* = 0.000034)) (Figure 6A–D).

Importantly, the proinflammatory potential of LPS administration was attenuated in clodronate-treated OHCs (IL-1β (*p* = 0.000280), CASP1 (*p* = 0.000114), NLRP3 (*p* = 0.000062), PYCARD (*p* = 0.000061)) (Figure 6A–D). Moreover, AMS21 seems to be microglia-dependent since we noticed the beneficial potential of this compound only in the absence of clodronate administration. In fact, OHCs treatment with clodronate inhibited the beneficial impact of AMS21 on the protein level of NLRP3 inflammasome-related factors in LPS-stimulated OHCs (IL-1β (*p* = 0.014792), CASP1 (*p* = 0.000054), NLRP3 (*p* = 0.000055), PYCARD (*p* = 0.000054)) (Figure 6A–D).

## 4. Discussion

In this study, we found that the FPR2 agonist AMS21 limits lactate dehydrogenase and nitric oxide release in OHCs stimulated by LPS. Moreover, AMS21 significantly attenuated LPS-evoked microglial marker expression, including *Cd40* and *Il-18*, while upregulating *Il-1Ra* expression, a negative regulator of the inflammatory response. The FPR2-dependent pro-resolving ability of AMS21 was related to the limitation of proinflammatory cytokines: IL-1β and partial IL-6 release in stimulated OHCs. Importantly, our results are the first to firmly point to the crucial role of FPR2 expressed by microglial cells in the anti-inflammatory activity of AMS21 since the depletion of microglial cells abolished this effect. Moreover, the molecular mechanism of the pro-resolving potential of AMS21 treatment in control OHCs was identified as the suppression of the NLRP3 inflammasome complex and the decrease in IL-1β release in the neuroinflammatory response.

Considering the inconclusive data related to the expression of FPR2 in CNS cells and the fact that microglial cells perform their functions in the brain mainly through interactions with other cells that are highly sensitive to the surrounding environment, in this study, we used an ex vivo model of hippocampal organotypic cultures. This model preserves the functional interactions between brain cells, and the influence of various biological brain components, including immunological components, on the studied effects has not been affected [55]. Moreover, this experimental model allowed us to demonstrate the role of microglial cells in the studied mechanisms related to FPR2 activation through simple pharmacological modulation.

Modulation of the resolution of inflammation (RoI) has been proposed as a new strategy to treat CNS disorders based on inflammation, and the FPR2 receptor has been recently identified as a target of pro-resolving agents [9,56,57]. Since endogenous FPR2 agonists (such as LXA4 and ATL-LXA4) are characterised by high chemical lability and poor bioavailability, the identification of “drug-like” FPR2 agonists is necessary [58,59]. During the last decade, intensive work has been carried out to identify compounds that could combine the favourable profile of endogenous ligands, including RoI, with favourable pharmacokinetic properties and high bioavailability. We identified the first series of FPR2 agonists based on an ureidopropanamide scaffold able to reduce the intracellular levels of proinflammatory mediators in rat primary microglial cell cultures stimulated with lipopolysaccharide (LPS) [31,60]. Furthermore, the compounds are stable to oxidative metabolism and have reasonable permeation rates in hCMEC/D3 cells, which are used as an in vitro blood barrier model. However, the most promising ligand, MR-39, produces beneficial in vitro effects in the micromolar range. Thus, this limits its use for in vivo preclinical studies because a high dosage would imply the risk of unpredictable and confounding off-target effects. Therefore, in a subsequent study, the FPR2 agonist potency of the ureidopropanamide derivatives was improved and led to the identification of compound AMS21, which was able to activate FPR2 at nanomolar concentrations [45]. In the present study, we evaluated the neuroprotective and anti-inflammatory effects of AMS21 in OHCs exposed to LPS, a well-accepted neuroinflammation model.

First, we demonstrated that AMS21 at nanomolar concentrations attenuated LDH release and NO production evoked by LPS. These data are consistent with our previous observations in the model of primary microglial cultures [45] and confirm the neuroprotective effects of AMS21 in the three-dimensional cell system that maintains neuronal–glial interactions, thus bringing us closer to preclinical studies in vivo. The endotoxin of Gram-negative bacteria is one of the most potent bacterial inducers of cytokine release, including proinflammatory cytokines such as IL-1β and IL-6 [61,62,63], and the gene expression of various other proinflammatory markers and factors. Consistent with these observations, in this study, LPS upregulated the expression of *Cd40*, *Il-1β*, *Il-18*, and *Il-6* in OHCs. Moreover, we observed an increased production of IL-1β and IL-6. AMS21 treatment abolished the stimulatory effect of LPS administration on the expression of proinflammatory markers, namely, *Cd40* and *Il-18*. Concurrently, the administration of this ligand diminishes the IL-1β and IL-6 levels. This effect was FPR2-dependent because it was counterbalanced by pre-treatment with the FPR2 antagonist WRW4 in LPS-stimulated OHCs. Therefore, the results reported in this study are in agreement with those obtained in other experimental models using the FPR2 agonist MR-39 [31,48,50]. Nevertheless, we showed that AMS21 in LPS-evoked cultures did not disturb the increase in the release of anti-inflammatory cytokines (TGF-β, IL-10), whose role as a “stop signal” in inflammatory processes is crucial. IL-10 exerts an anti-inflammatory response at least in part by regulating IL-1β production [50]. LPS specifically activates IL-10, triggering the induction of IL-1β secretion, whose level, together with the amount of pro-IL-1β, determines the final level of IL-1β [64]. Since we found that the antagonist WRW4 inhibits the synthesis of IL-10 and TGF-β, it can be suggested that the activation of FPR2 by AMS21 plays an indirect role in maintaining a proper balance between pro- and anti-inflammatory cytokines, thus contributing to the regulation of RoI and return to homeostasis after LPS-induced immune activation. Consistent with this hypothesis, we found that AMS21 upregulates *Il-1Ra* in OHCs. This protein inhibits the IL-1β receptor and is an essential factor in regulating IL-1-mediated inflammation [65]. Interestingly, some data also indicate the positive synergistic effects of IL-1Ra with TGF-β and IL-10 cytokines [66], pointing to the potential new mechanism for promoting RoI by AMS21, but this observation requires further verification.

As we have shown that the beneficial activity of AMS21 is mainly associated with the inhibition of the proinflammatory response related to the IL-1 family, we assessed the molecular mechanism underlying the observed anti-inflammatory and pro-resolving effects of AMS21, focusing our attention on the NLRP3 inflammasome. NLRP3 inflammasome activation leads to the release of the active cytokines IL-1β and IL-18 in a model of neuroinflammation in OHCs [50]. Indeed, IL-1β is biologically inactive and must be cleaved and transformed into its biologically active form by the enzymatic activity of caspase-1 [17,54,67]. Therefore, upon activation, NLRP3 nucleates PYCARD helical clusters through PYD-PYD interactions. The oligomerised PYCARD CARDs then form the platform for caspase-1 CARD to nucleate into filaments, which, in turn, activates caspase-1 [54,68]. Unique among inflammasomes, NLRP3 requires a priming step for canonical activation. In experimental practice, LPS treatment is able to induce NLRP3 expression. In the next step, NLRP3 inflammasome assembly leads to inflammasome activation [54]. We found that LPS stimulation upregulated *Nlrp3* and *Pycard* subunits expression in OHCs. AMS21 attenuated the impact of endotoxin on the mRNA expression of the *Nlrp3* subunit. Furthermore, we demonstrated that the LPS-evoked increase in CASP-1, NLRP3, and PYCARD levels in OHCs was inhibited by AMS21 administration. Notably, this beneficial effect of AMS21 was abolished by WRW4 pre-treatment, thus confirming that the effect is FPR2-mediated. This observation suggests a potential interaction between FPR2 activation and the suppression of the canonical NLRP3 inflammasome pathway in OHCs. Interestingly, the anti-inflammatory activity of AMS21 was observed at nanomolar concentrations, providing grounds for further research on the pro-resolving potential of this compound in vivo. Moreover, in our previous study, we observed that the levels of caspase-1 in OHCs from KO FPR2 mice were higher than those in OHCs obtained from WT mice, suggesting enhanced cleavage of pro-IL-1β into active IL-1β and a consequential increase in the level of IL-1β in OHCs from KO FPR2 mice. Additionally, MR-39 was able to diminish caspase-1 activation only in KO FPR2 mice [50].

To date, most of the data indicate the expression of FPR2 on microglial cells and the unique ability of these receptors to differentiate responses based on the structure of its ligands following the described agonist bias [69,70]. Nevertheless, other authors point to the expression of FPR2 on other brain cells, including neurons, astrocytes, and even oligodendrocytes [35,71]. To shed more light on this still controversial issue, in the following research stage, we used a method of eliminating microglial cells at the stage of OHCs establishment. To date, experimental approaches to characterise microglia’s functional receptors and repertoire have relied on pharmacological and genetic methodologies. Among these approaches is the use of bisphosphonate clodronate, which deactivates cells belonging to the monocyte lineage [72,73]. Clodronate is a first-generation bisphosphonate that exerts an effect on the brain’s peripheral macrophages and perivascular cells [73,74]. Simultaneously, the most spectacular direct effect of clodronate on microglial cells involves inhibiting proliferation [75]. Accordingly, we have shown, using anti-IBA1 staining, that clodronate eliminated microglial cells, while stimulation with bacterial endotoxin did not increase the IBA1 protein level. At the same time, using confocal microscopy, we showed the colocalisation of FRP2 with IBA1-positive cells, which was absent after microglia depletion in OHCs. Nevertheless, the measurement of FPR2 fluorescence demonstrated no complete decrease in FPR2 level, which suggests the presence of this receptor in OHCs after clodronate treatment, although it was significantly reduced. These observations indicate the localisation of FPR2 in other cells in OHCs, probably astrocytes and/or neurons [76].

Moreover, the various FPR2-mediated activities of microglial cells were affected by clodronate treatment. We demonstrated that clodronate diminished IL-1β release in OHCs compared to control cultures, but only after LPS treatment. Other studies have reported that TNF-β and IL-6 production is also limited after immune stimulation in microglia-depleted OHCs [75]. According to data on RAW 264 macrophages, proinflammatory cytokines and NO production decreased following clodronate treatment [77]. The mechanisms of action of bisphosphonates have not been fully clarified. In 1997, Frith et al. [78] showed that clodronate can be metabolised, and the metabolites inhibit the DNA-binding activity of NF-κB and the production of proinflammatory cytokines. In the present study, we demonstrated for the first time that NLRP3 pathway activation is attenuated, including caspase-1 level and other protein subunits, following clodronate administration in OHCs.

Additional intriguing points raised in our study are that microglia depletion completely abolished the anti-inflammatory potential of AMS21 administration observed in control OHCs. It may be hypothesised that the depletion of FPR2, which is located mainly on microglial cells, significantly hampered the possibility of the ligand to interact with its own receptor, thus limiting its beneficial effect. Interestingly, in our previous study, we observed that the beneficial impact of the structurally related FPR2 agonist MR-39 was also limited to the suppression of microgliosis but not astrogliosis in an *in vivo* model of Alzheimer’s disease [48].

Although the data indicate that clodronate mainly affects microglia, the possibility that the impact of clodronate on OHCs was also induced indirectly via other glial cells on which FPR2 was preserved cannot be ruled out. Accordingly, clodronate in adult mice leads to microenvironment changes that decrease neuronal markers and blood vessel integrity [79]. In contrast, clodronate administration in vitro improves the purity of astrocytes and increases the postsynaptic current frequency in OHCs [80,81]. Moreover, in microglia-eliminated cultures, astrocytes produced IL-6, while IL-1β followed the activation of the JAK/STAT3 pathway [80]. Recent advances in genomics and multiomics have yielded novel insight into astrocyte reactivity, in which astrocytes undergo a broad spectrum of morphological, molecular and functional changes to become reactive astrocytes [82,83]. The transformation to reactive phenotypes involves a variety of molecular regulators and signalling pathways [84]. Nevertheless, despite many efforts, the mechanism of astrocyte activation, such as their response to TLR ligands, including LPS, remains highly debated [85]. Among others, CD14 was found in astrocytes, which serve as a high-affinity receptor for LPS and correspond to the interaction with TLR4 activated by this endotoxin [85]. Moreover, rodent astrocytes have been shown to be highly sensitive to IL-1β, while the types of inflammatory genes induced by this cytokine resemble those of LPS-activated microglia, suggesting that astrocytes are capable of mounting potent immune responses [86,87,88]. Therefore, it can be suggested that the IL-1β level observed in OHCs treated with clodronate, at least in part, has an astrocytic origin.

The next intriguing observation from the present study is NLRP3 activation in clodronate-depleted OHCs. Although some data suggest that the NLRP3 inflammasome is limited to microglia but not astrocytes [89], increasing evidence supports the presence of the NLRP3 inflammasome in other brain cells, such as oligodendrocytes and astrocytes, in pathological conditions and some disease models [90,91]. Our research showed the activation of not only CASP1 but also of the remaining two NLRP3 subunits evoked by LPS in microglia-depleted OHCs. This finding strongly suggested the presence of the NLRP3 inflammasome in astroglia. It should be emphasised that other caspases can also activate NLRP3 by non-classical and/or alternative activation pathways. An emerging body of research has supported the role of caspase-4, caspase-5, and caspase-11 in regulating caspase-1 activation and inducing the inflammatory form of cell death called pyroptosis [92,93]. Moreover, the role of caspase-12, which is expressed by astrocytes and is important for caspase-1 and NLRP3 activation, should also be considered [94].

## 5. Conclusions

The FPR2 receptor is a versatile transmembrane receptor that recognises a wide variety of chemically diverse endogenous ligands. This creates a unique opportunity to switch from a pro- to an anti-inflammatory activation profile of this receptor in the brain. In this context, our study provides new data on the molecular mechanisms underlying the anti-inflammatory and pro-resolving properties of our second-generation FPR2 agonist AMS21 in an ex vivo model of an experimentally induced neuroinflammatory response. Our data showed that AMS21 modulates the proinflammatory response related to the IL-1 family through different mechanisms that include the modulation of inflammasome NLRP3 assembly and the upregulation of the IL-1Ra protein. These data propose a new mechanism for the pro-resolving effect of AMS21 that will be further studied in detail in future studies. Finally, our findings suggest a crucial role of microglial cells and the FPR2 receptor located on these glial cells in mediating the anti-inflammatory response of compound AMS21.

Therefore, these studies have significant implications for the translation of FPR2 activation and modulation of downstream signalling as a potential therapeutic agent for inflammatory pathologies.

## Figures and Tables

**Figure 1 cells-12-02570-f001:**
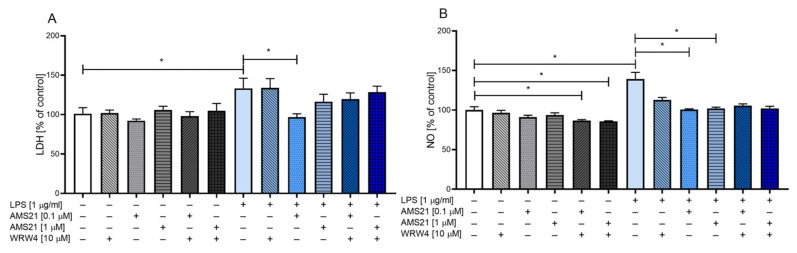
The effect of WRW4 and AMS21 treatment on lactate dehydrogenase (LDH) (**A**) and nitric oxide (NO) (**B**) release in LPS-stimulated OHCs. Cultures were pre-treated with the FPR2 antagonist WRW4 (10 μM) for 30 min. Afterwards, OHCs were treated with two doses of AMS21 (0.1 μM and 1 μM) for one hour and then stimulated with lipopolysaccharide (LPS; 1 μg/mL) for 24 h. Control groups were treated with an appropriate vehicle. The data are presented as the mean percentage ± SEM of the control of independent experiments, *n* = 4–6 in each experiment. Statistical analysis was carried out using two-way analysis of variance (ANOVA) with the Duncan post hoc test to assess the differences between the treatment groups. Significant differences are indicated by * *p* < 0.05. LDH—lactate dehydrogenase; NO—nitric oxide; LPS—lipopolysaccharide.

**Figure 2 cells-12-02570-f002:**
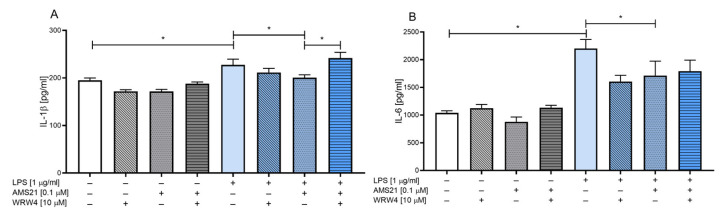
The impact of WRW4 and AMS21 on the release of proinflammatory IL-1β (**A**) and IL-6 (**B**) cytokines in LPS-stimulated OHCs. Cultures were pre-treated with the FPR2 antagonist WRW4 (10 μM) for 30 min. Afterwards, OHCs were stimulated with AMS21 (0.1 μM) for one hour and then with lipopolysaccharide (LPS; 1 μg/mL) for 24 h. Control groups were treated with the appropriate vehicle. The data are presented as the mean ± SEM of independent experiments, *n* = 4–8 in each experiment. Statistical analysis was carried out using two-way analysis of variance (ANOVA) with the Duncan post hoc test to assess the differences between the treatment groups. Significant differences are indicated by * *p* < 0.05. IL-1β—interleukin 1β; IL-6—interleukin 6; LPS—lipopolysaccharide.

**Figure 3 cells-12-02570-f003:**
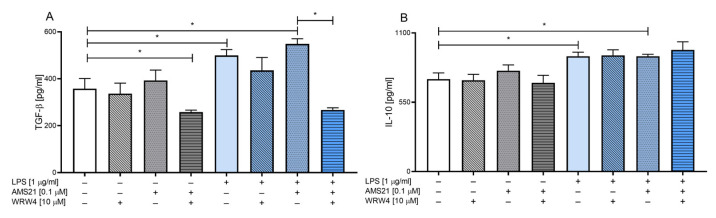
The impact of WRW4 and AMS21 on the release of anti-inflammatory TGF-β (**A**) and IL-10 (**B**) cytokines in LPS-stimulated OHCs. Cultures were pre-treated with the FPR2 antagonist WRW4 (10 μM) for 30 min. Afterwards, OHCs were stimulated with AMS21 (0.1 μM) for one hour and then with lipopolysaccharide (LPS; 1 μg/mL) for 24 h. Control groups were treated with the appropriate vehicle. The data are presented as the mean ± SEM of independent experiments, *n* = 4–8 in each experiment. Statistical analysis was carried out using two-way analysis of variance (ANOVA) with the Duncan post hoc test to assess the differences between the treatment groups. Significant differences are indicated by * *p* < 0.05. TGF-β—transforming growth factor β; IL-10—interleukin 10; LPS—lipopolysaccharide.

**Figure 4 cells-12-02570-f004:**
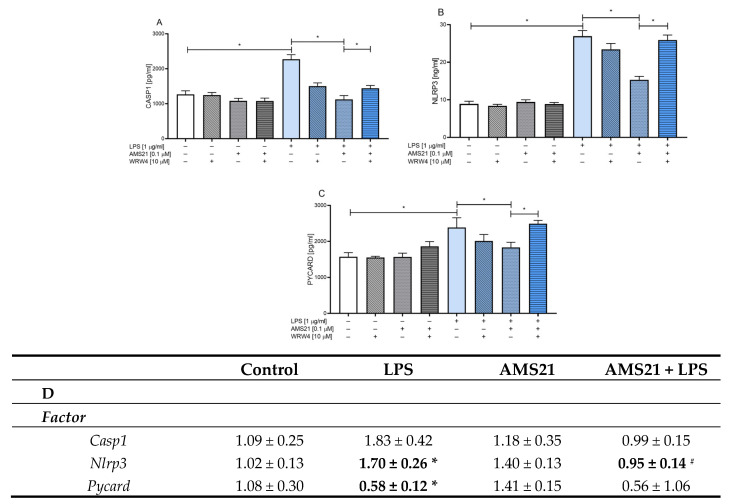
The impact of WRW4 and AMS21 on the protein levels of CASP1 (**A**), NLRP3 (**B**), and PYCARD (**C**) and mRNA expression (**D**) in LPS-stimulated OHCs. Cultures were pre-treated with WRW4 (10 μM) for 30 min. Afterwards, OHCs were administered AMS21 (0.1 μM) for one hour and lipopolysaccharide (LPS; 1 μg/mL) for 24 h. Control groups were treated with the appropriate vehicle. The data are presented as the mean ± SEM of independent experiments (**A**–**C**), *n* = 6–8 in each experiment and as the average fold change ± SEM (**D**), *n* = 3–6 in each experiment. Statistical analysis was carried out using two-way analysis of variance (ANOVA) with the Duncan post hoc test to assess the differences between the treatment groups. Significant differences are indicated by * *p* < 0.05 (**A**–**C**), and * *p* < 0.05 control compared to the LPS group; ^#^
*p* < 0.05 LPS compared to the AMS21 + LPS group (**D**). CASP1—caspase 1; NLRP3—nod-like receptor pyrins 3; PYCARD—apoptosis-associated speck-like protein containing a caspase recruitment domain; LPS—lipopolysaccharide.

**Figure 5 cells-12-02570-f005:**
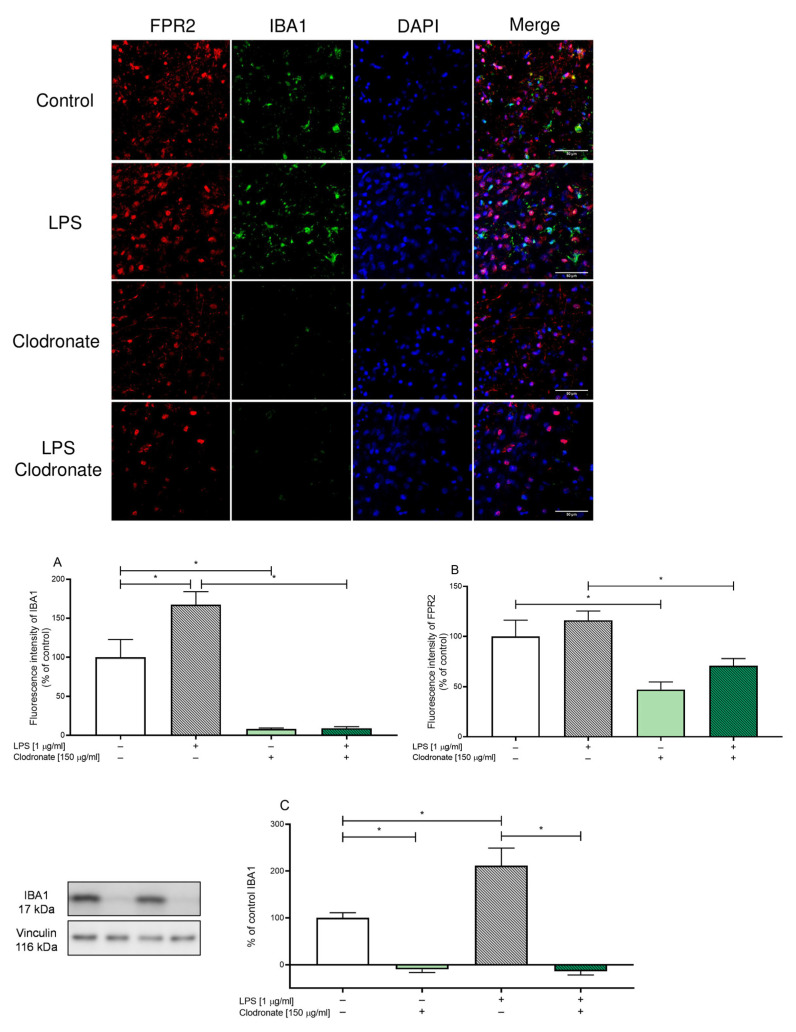
Representative fluorescence images of OHCs obtained using confocal microscopy (**A**,**B**) and the protein level of IBA1 (**C**) after LPS and clodronate administration. Microglia depletion was performed on the 1st DIV using clodronate (150 μg/mL). On the 7th DIV, cultures were treated with lipopolysaccharide (LPS; 1 μg/mL) for 24 h. Control groups were treated with the appropriate vehicle. Fluorescence intensity data are derived for IBA1 (**A**) and FPR2 (**B**). Nuclei appear in blue Hoechst 33342, FPR2 was labelled using AlexaFluor^®^ 647 in red and IBA1 was labelled using AlexaFluor^®^ 555. The data are presented as the mean ± SEM of independent experiments, *n* = 6–9 for fluorescence images, and *n* = 4 for Western blot analysis. Statistical analysis was carried out using two-way analysis of variance (ANOVA) with the Duncan post hoc test to assess the differences between the treatment groups. Significant differences are indicated by * *p* < 0.05. Scale bar: 50 μm is located at the bottom right corner of each image.

**Figure 6 cells-12-02570-f006:**
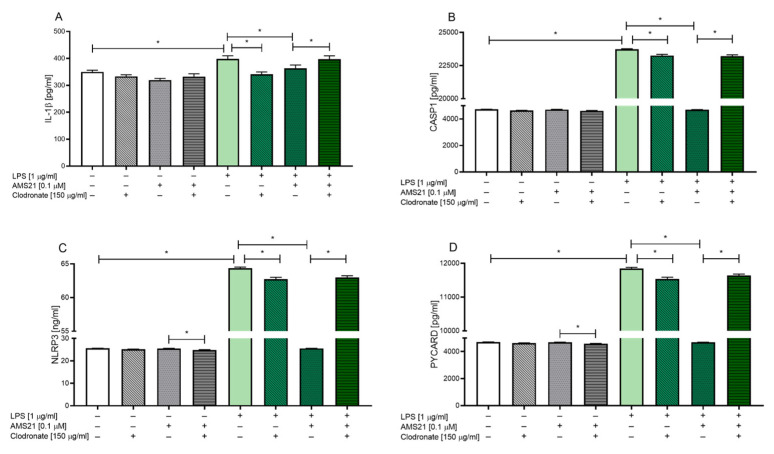
The impact of clodronate and AMS21 on the protein levels of IL-1β (**A**), CASP1 (**B**), NLRP3 (**C**), and PYCARD (**D**) in LPS-stimulated OHCs. Microglia depletion was performed on the 1st DIV using clodronate (150 μg/mL). On the 7th DIV, cultures were stimulated with AMS21 (0.1 μM) for one hour and then with lipopolysaccharide (LPS; 1 μg/mL) for 24 h. Control groups were treated with the appropriate vehicle. The data are presented as the mean ± SEM of independent experiments, *n* = 7–8 in each experiment. Statistical analysis was carried out using two-way analysis of variance (ANOVA) with the Duncan post hoc test to assess the differences between the treatment groups. Significant differences are indicated by ** p <* 0.05. CASP1—caspase 1; NLRP3—nod-like receptor pyrins—3; PYCARD—apoptosis-associated speck-like protein containing a caspase recruitment domain (CARD); LPS—lipopolysaccharide.

**Table 1 cells-12-02570-t001:** The impact of AMS21 on the mRNA expression of proinflammatory (*Cd40*, *Cd68*, *Il-1β*, *Il-6*, *Il-18*) (A) and anti-inflammatory (*Igf-1*, *Il-1Ra*, *Tgf-β*) (B) genes in LPS-stimulated OHCs. OHCs were stimulated with AMS21 (0.1 μM) for an hour and subsequently with lipopolysaccharide (LPS; 1 μg/mL) for 24 h. Control groups were treated with the appropriate vehicle. The data are presented as the average fold change ± SEM of independent experiments, *n* = 3–6 in each experiment. Statistical analysis was performed using two-way analysis of variance (ANOVA) with the Duncan post hoc test to assess the differences between the treatment groups. Significant differences are indicated by ** p* < 0.05 control compared to the LPS group; *^#^ p* < 0.05 LPS compared to the AMS21 + LPS group.

	Control	LPS	AMS21	AMS21 + LPS
**A**				
** *Proinflammatory factors* **				
*Cd40*	1.04 ± 0.18	**2.83 ± 1.51 ***	1.02 ± 0.25	**0.96 ± 0.14 ^#^**
*Cd68*	1.06 ± 0.18	1.07 ± 0.39	0.92 ± 0.20	0.94 ± 0.16
*Il-1β*	1.06 ± 0.15	**27.05 ± 3.70 ***	1.02 ± 0.18	**27.58 ± 1.90 ***
*Il-6*	1.07 ± 0.27	**3.57 ± 0.33 ***	3.23 ± 1.19	3.42 ± 0.24
*Il-18*	1.05 ± 0.18	**2.52 ± 0.86 ***	1.04 ± 0.29	**0.97 ± 0.18 ^#^**
**B**				
** *Anti-inflammatory factors* **				
*Igf-1*	1.04 ± 0.15	**0.33 ± 0.02 ***	1.06 ± 0.18	0.47 ± 0.10
*Il-1Ra*	1.06 ± 0.28	**7.18 ± 1.63 ***	1.59 ± 0.25	**11.74 ± 1.98 ^#^**
*Tgf-β*	1.03 ± 0.16	0.45 ± 0.14	1.15 ± 0.21	0.84 ± 0.12

## Data Availability

All data supporting the conclusions of this manuscript are provided in the text, figures, and tables.

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
