# Peer review of "Microglia Depletion Attenuates the Pro-Resolving Activity of the Formyl Peptide Receptor 2 Agonist AMS21 Related to Inhibition of Inflammasome NLRP3 Signalling Pathway: A Study of Organotypic Hippocampal Cultures"

_cells, 2023, doi:10.3390/cells12212570_

Round 1
Reviewer 1 Report
Comments and Suggestions for Authors
To gain my final endorsement, please consider the following points:
1. Please make the Abstract shorter and more narrative, rather than an extensive re-cap of the entire manuscript.
2. Introduce and discuss further the AMS21 specificity for FPR2 over other FPRs and receptors.
3. Introduce and discuss the expression and changes of it in different cell types in the brain both at the physiological and pathological levels.
4. Displayed changes on LDH and NO assays seem statistically significant but rather small and misled by the expression of the result using %. Please include standard curves for these assays and express the results as n° dead cells (LDH) and microM (NO2-). What are the basal levels in your control cultures?. I agree with the discussion in line 629, but the current study promises the evaluation of AMS21 effects in a more complex culture model where cellular interactions are more preserved. So, How do they operate then?
5. The cell type releasing LDH and producing NO is not addressed at all. Are the same microglia? exclusively? Please include evidence identifying the source of these molecules.
6. LPS induction of IL1 and TGFB is again modest. Please consider adjusting the concentration of and/or exposition time to LPS in order to produce a clear proinflammatory reaction in the culture system.
7. It confuses me that the levels of different inflammatory mediators seem affected by AMS21 at the protein level but not at the mRNA level. For instance, but not only, LPS-induced levels of active IL1B are decreased by AMS21 while its mRNA levels remain high in the same condition.
Author Response
First of all, we would like to express our sincere gratitude to the Reviewer for the constructive comments on the initial version of our manuscript. Please find our responses to the raised issues hereafter.
Point 1. Please make the Abstract shorter and more narrative, rather than an extensive re-cap of the entire manuscript.
Response 1: Thank you very much for this valuable observation. As suggested by the reviewer, the abstract has been redefined in a more narrative form in the revised version of our paper.
Point 2. Introduce and discuss further the AMS21 specificity for FPR2 over other FPRs and receptors.
Response 2: Thank you for this suggestion. As reported in Mastromarino et al (ref 43 of the manuscript) AMS21 is more than 10-fold selective over FPR1. This information is clearly stated in the revised version of our manuscript. Unfortunately, as for the selectivity over other receptors, it is difficult to establish a meaningful selectivity panel considering that the inflammatory response induces the over-expression of numerous receptors. In the present study plan, we included the co-administration of the selective antagonist WRW4 as a way to assess the specificity of the effects induced by the treatment with AMS21. As the observed anti-inflammatory and pro-resolving effects of AMS21 were, at least in part, counterbalanced by the WRW4 treatment, it is reasonable that AMS21 elicits its effects mainly through interaction with FPR2. Also, this aspect is now discussed in the revised version of our manuscript.
Point 3. Introduce and discuss the expression and changes of it in different cell types in the brain both at the physiological and pathological levels.
Response 3: Thank you for bringing this issue to our attention. We didn't describe in detail information about the expression level and function of FPR2 in various types of cells in the central nervous system because this topic has been the subject of previous our manuscripts (DOI: 10.2174/1570159X17666191019170244 (Trojan et al. 2020), DOI: 10.2174/1570159X20666220913155248 Trojan et al.2022, https://doi.org/10.1007/s43440-021-00271-x Tylek et al. 2021, ). Moreover, its expression under physiological conditions in microglia, astrocytes, and neurons was demonstrated using immunohistochemistry in recently published work (https://doi.org/10.1021/acschemneuro.3c00525 (Tylek et al., 2023). It should be noted that the also other studies have focused on examining the expression of the FPR2 receptor both under physiological and pathological conditions, namely microglia stimulation using lipopolysaccharide (10.3390/cells10092373 (Tylek et al., 2021)), microglia and astrocytes in Alzheimer's disease research using Aβ (https://doi.org/10.1111/j.1471-4159.2010.06637.x (Brandenburg et al., 2010)), or even subarachnoid hemorrhage (SAH) models applied to primary neuron and microglia cultures (https://doi.org/10.1186/s12974-020-01918-x (Liu et al., 2020)).
Nevertheless, according to the reviewer's suggestion in the revised version of the manuscript, we completed in a cumulative way the relevant citations in the introduction section: "As demonstrated in previous scientific research in the brain, FPR2 is expressed mainly in microglia cells, neurons, and astroglia not only in physiological but also pathological conditions”.
Point 4. Displayed changes on LDH and NO assays seem statistically significant but rather small and misled by the expression of the result using %. Please include standard curves for these assays and express the results as n° dead cells (LDH) and microM (NO2-). What are the basal levels in your control cultures? I agree with the discussion in line 629, but the current study promises the evaluation of AMS21 effects in a more complex culture model where cellular interactions are more preserved. So, How do they operate then?
Response 4: Thank you for your valuable suggestions. Lactate dehydrogenase (LDH) assay is a method to quantify cytotoxicity based on the measurement of LDH activity released from damaged cells. Griess reaction was used to detect and quantify the level of nitric oxide (NO) released to the culture medium in OHC. Although OHC are extremely valuable method for studying neuron-glia interactions, has certain limitations. One of the most characteristic limitations of OHC without a doubt is differences between groups obtained in independent experiments including differences between control groups. Accordingly, the levels of LDH and NO in OHC obtained from independent experiments are expressed as a percentage of appropriate control for a given experimental condition. To be precise: we performed the test in accordance with the manufacturer's instructions to determine the percentage of cytotoxicity, we had to calculate the average absorbance values of the background from samples and controls, and then substitute the resulting values in the equation described in the manual. Because there is some difference in the level of cytotoxicity or NO release between cultures (the direction is the same, but sometimes the level is different), in order to be able to compare them, we converted each experiment into % of control.
We agree with the opinion of the reviewer that OHC is a more complex culture model where cellular interactions are preserved. In fact, the presence of neuronal and glial cells was demonstrated in hippocampal cultures, also in our previous studies (Tylek et al. 2023). In the present studies, we demonstrated that the elimination of microglia, the main source of FPR2 expression, significantly reduces the pro-resolving potential of AMS21, which gives us reason to believe that microglial FPR2 is the main target for this ligand. Nevertheless, FPR2 is also expressed on neurons and, in special cases (inflammatory process, damage), on astrocytes; therefore, the interactions taking place in the brain between these cells and microglia allow us to suggest that these receptors also mediate the action of AMS21, although to a lesser extent.
Point 5. The cell type releasing LDH and producing NO is not addressed at all. Are the same microglia? exclusively? Please include evidence identifying the source of these molecules.
Response 5: Thank you for raising this issue to our attention. Lactate dehydrogenase (LDH) is an essential enzyme of the anaerobic metabolic pathway. It belongs to the class of oxidoreductases. The enzyme's function is to catalyze the reversible conversion of lactate to pyruvate with the reduction of NAD+ to NADH and vice versa. The enzyme is present in various organisms, including animals and plants (Chen 2014, Kumar et al., 2018). It is ubiquitously present in all tissues and is an important checkpoint for gluconeogenesis and DNA metabolism. A species-wide analysis of LDH demonstrates its well-preserved structure with only a few changes in the amino acid sequence across species. The structural similarity with slight amino acid changes provides a logical platform for designing functional molecules to modulate the catalytic potential and expression of the enzyme (Farhana and Lappin, 2023). LDH is rapidly released into the cell culture supernatant when the plasma membrane is damaged, a key feature of cells undergoing apoptosis, necrosis, and other forms of cellular damage. Since in our study, the LDH level was measured in the medium, it is not possible to assess which cells in the OHC were its source in detail.
Brain-derived NO is mainly produced by the neuronal form of NOS (nNOS) through a reaction that converts L-arginine and oxygen into citrulline and NO. The nNOS enzyme is one of the three homologous isoforms of NOS, with the other two being endothelial NOS (eNOS) and inducible NOS (iNOS). This iNOS promoter activity has been shown to be restricted to neurons in the healthy brain. In contrast, inflammatory processes are seen to induce its expression in microglia but not in neurons (Bechade et al., 2014). In the mature brain, even though iNOS activity has been shown to have beneficial effects when the brain gets injured (Sinz et al., 1999), iNOS is usually associated with pathophysiological situations. Therefore, in OHC stimulated with bacterial endotoxin, microglia are the primary sources of NO. Because of its numerous physiological and pathophysiological roles, NO can act as a “double-edged sword”, and it has been demonstrated that clarification of the dual effect of NO might have implications in researching new potential pharmacological tools. Additionally, the ubiquity of NO in the CNS implies that drugs designed to modify the biological activity of NO may have distinct effects. In the present study, the measurement of LDH and NO levels shown in Fig. 1 was used as a preliminary study to select the most effective doses of compounds. Moreover, the obtained result will allow us to assess whether the compounds are not toxic at the concentrations used. For the same purpose, measurement of LDH and NO release (to assess NO release) after stimulation with toxic agents was commonly used previously (e.g. Jantas et al. 2020, Tylek et al. 2021).
Point 6. LPS induction of IL1 and TGFB is again modest. Please consider adjusting the concentration of and/or exposition time to LPS in order to produce a clear proinflammatory reaction in the culture system.
Response 6. Thank you for drawing our attention to this matter. The model of induction immunological activity in OHC using LPS at a dose of 1μg/ml for 24 hours is one of the well-established ex vivo model of inflammation in the CNS. Numerous publications (https://doi.org/10.1021/acschemneuro.3c00525 (Tylek et al. 2023); .https://doi.org/10.3390/cells10061524 (Trojan et al. 2021) confirm that an increased level of both pro- and anti-inflammatory cytokines is a characteristic element that occurs after LPS stimulation. In the current work, although the levels of IL-1β and TGF-β may not seem to be significantly increased after LPS stimulation, the difference between the control group and the LPS group is high enough to achieve statistical significance. However, it should underlined that OHC are primary cultures directly derived from animals and that differences in inflammatory activation may occur in each individual experiment performed. Nevertheless, the induction of statistically significantly increased levels of mentioned cytokines by LPS is observed every time.
Point 7. It confuses me that the levels of different inflammatory mediators seem affected by AMS21 at the protein level but not at the mRNA level. For instance, but not only, LPS-induced levels of active IL1B are decreased by AMS21 while its mRNA levels remain high in the same condition.
Response 7: Thank you for bringing up this issue. The inconsistent impact of AMS21 on this parameter may result from alterations in the regulation of various stages of mRNA expression, starting with changes in chromatin conformation, gene activation in response to external stimuli, and control of the transcription and translation (Doma & Parker, 2007; Houseley & Tollervey, 2009). Another possible reason for this phenomenon is the disruption in the mRNA's nuclear retention, an essential mechanism for maintaining the dynamic balance between de novo transcription and protein translation (Mazille et al., 2022). Moreover, IL-1β is a potent pro-inflammatory cytokine and thus subject to tight multilayered control. The binding of IL-1β to the IL-1 receptor triggers a MyD88-dependent kinase cascade that culminates in activating several transcription factors, such as NF- κB and AP1, and subsequent gene expression changes. IL-1 signaling is regulated at its level of bioavailability, and the process is shown to involve transcription, post-translational cleavage, and liberation from the cell. In contrast, studies addressing the regulation of IL-1 cytokines at the level of their mRNA stability are scarce. It has been found that mRNA decay is context-dependent and mainly associated with stabilizing intrinsically unstable, inflammation-induced mRNAs. It is likely that LPS stimulation also modulates these processes in a context-dependent manner, influencing the observed changes in the assessed parameters following ligand administration. While the changes induced by various factors in gene expression can have significant consequences and provide valuable insights into cellular processes under investigation, the biological effect of the tested compound ultimately depends on the levels of the proteins involved after considering post-translational changes.
Once again, we would like to express our gratitude for the time spent reviewing our manuscript and all the insightful remarks. We do hope that the corrections made in the revised version of the article, based on the suggestions of the Reviewers, improved the quality of our paper and that our responses clarified all of the questions.
Reviewer 2 Report
Comments and Suggestions for Authors
In the article entitled: “Microglia Depletion Attenuates the Pro-Resolving Activity of the Formyl Peptide Receptor 2 Agonist AMS21 Related to Inhibition of Inflammasome NLRP3 Signalling Pathway: A Study of Organotypic Hippocampal Cultures”, the authors Study the role of FPR2 in microglia in mediating the anti-inflammatory response by means of a novel agonist of this receptor. The relevance of this study is highlighted by the use of an antagonist of FPR2 and a depletor of microglia: in this way it have been possible to verify the effective involvement of FPR2 in anti inflammatory responses observed in hippocampal slices.
Although the article is quite clear and well written there are some points that need to be addressed before publishing:
164: The authors have to explain why initially they have selected the two doses of 0.1 μM and 1 μM. It’s a ten fold difference, do they have previous data that justify this? Why not to use also an intermediate dilution, such as 0.5 uM?
252: Was IF staining performed straight on 350um slices? Ab penetration as well as slide observation are quite difficult on 350 um thick sections. In the protocol cited 100/150 um thick ones were used, indeed.
347, Results section 3.3: IL-10 release seems not to be related by AMS21 because the blocking of FPR2 with the antagonist does not change its levels. However, it should be that its expression it’s modified: why haven’t be evaluated the mRNA expression of IL-10 and TGF, together with that of other anti inflammatory mediators? (in results section 3.4)
442 (figure 5, IF images): comparing Iba1 stainings, it seems that there are no differences in microglia morphology between control and LPS treated cells. It’s well known that following activation microglia morphology changes dramatically. I suggest to add higher magnification images showing microglia morphology.
Comments on the Quality of English Languagegood
Author Response
First of all, we would like to express our sincere gratitude to the Reviewer for the constructive comments on the initial version of our manuscript. Please find our responses to the raised issues hereafter.
Point 1 The authors have to explain why initially they have selected the two doses of 0.1 μM and 1 μM. It’s a ten fold difference, do they have previous data that justify this? Why not to use also an intermediate dilution, such as 0.5 uM?
Response 1: Thank you for raising this critical issue. The studied doses were selected according to the data reported in Mastromarino et al. (2022). In that paper, the effect of AMS21 on LPS-induced production of TNF-α and IL-1β was tested in mouse N9 microglia cells in a wide range of concentrations. We found that the effect of the compound at 0.5 μM and 1 μM were nearly comparable. Therefore, in organotypic cultures, we chose to test the compound only in one of these concentrations, namely at 1 μM, and compare these effects with a 10-fold lower concentration, namely 0.1 μM. The rationale for using an agonist at such a low concentration to activate FPR2 was the observation that the endogenous FPR2 agonist LXA4 also exerts a pro-resolving effect at concentrations comparable to those used (Tylek et al., 2021).
Point 2 Was IF staining performed straight on 350um slices? Ab penetration as well as slide observation are quite difficult on 350 um thick sections. In the protocol cited 100/150 um thick ones were used, indeed.
Response 2: Thank you very much for this precious observation. Indeed, our sections were 350 µm thick. Therefore, we included information in the “materials and methods” section that the Gogolla et al. (2006) protocol has been slightly modified. The purpose of staining sections with a thickness of 350 µm, and not 100-150 µm as proposed in the protocol, was to standardize OHC cultures because we used 350 µm sections for biochemical studies. The penetration of the antibody through the section with a thickness of 350 µm was, at the beginning, one of our issues. Therefore, we used the 1:50 dilution for primary antibodies and 1:300 for secondary antibodies, and the incubation time with antibodies was slightly longer than in the protocol. A similar protocol using 350-thick sections for IF testing was also used in our previously published work by Bryniarska-Kubiak et al., (2023).
Point 3 Results section 3.3: IL-10 release seems not to be related by AMS21 because the blocking of FPR2 with the antagonist does not change its levels. However, it should be that its expression it’s modified: why haven’t be evaluated the mRNA expression of IL-10 and TGF, together with that of other anti-inflammatory mediators? (in results section 3.4)
Response 3: Thank you for raising this important issue. In fact, AMS21 treatment decreased induced by LPS IL-10 release. However, this effect at the protein level was not modulated by WRW4. Nevertheless, we fully agree with the reviewer that our observations do not allow us to exclude a potential FPR2-dependent effect of AMS21 on Il-10 mRNA expression. Recent data indicate diverse mechanisms of cytokine gene expression regulation and their secretion level by various factors. These differences may be related to alterations in the regulation of various stages of mRNA expression, starting with changes in chromatin conformation, gene activation in response to external stimuli, and control of the transcription and translation (Doma & Parker, 2007; Houseley & Tollervey, 2009). Another possible reason for this phenomenon is the mRNA's nuclear retention disruption, an essential mechanism for maintaining the dynamic balance between de novo transcription and protein translation (Mazille et al., 2022). Moreover, it has been found that mRNA decay is context-dependent. Nevertheless, although the changes induced by various factors in gene expression can have significant consequences and provide valuable insights into cellular processes under investigation, the biological effect of the tested compound ultimately depends on the levels of the proteins involved after considering post-translational changes.
In the case of measuring the level of TGF-β, we showed that AMS21 treatment maintains the high level of TGF-β induced by LPS, and the antagonist administration reduces it in the absence of such changes at the gene expression level. Thus, it can be suggested that the pro-resolving ability of AMS21 is related to solid suppression of pro-inflammatory cytokines release and an attempt to maintain a high level of anti-inflammatory cytokines, and thus the balance between pro- and anti-inflammatory cytokines contributing to the RoI regulation after LPS-induced immune activation.
Point 4 (figure 5, IF images): comparing Iba1 staining, it seems that there are no differences in microglia morphology between control and LPS-treated cells. It’s well known that following activation microglia morphology changes dramatically. I suggest to add higher magnification images showing microglia morphology.
Response 4: Thank you for bringing up this issue. We agree with the reviewer that the confocal microscopy image submitted initially did not reflect the dramatic changes in the morphology of microglia after LPS stimulation. Nevertheless, the effect of immunostimulation on microglial cells is very diverse. Data indicate that they change the morphology of microglia in a brain region-specific manner (Schwarz J.M., et. al., J Neurochem 2012 doi: 10.1111/j.1471-4159.2011.07630. x). There are differences in the deramification of microglia, the retraction of microglial processes, and microglia adopting an amoeboid shape after LPS treatment. It is observed that differences in microglial number and morphology were also accompanied by differences in the gene expression of cytokines, which indicate that microglial morphology, and likely their physiological state, are affected by LPS. Nevertheless, considering the reviewer's suggestion and in order to better visualize microglial activity, in the revised version of our manuscript, we included a different image of LPS-evoked OHC in Figure 5. Moreover, we took higher magnification to individual cells in the control and LPS-stimulated groups and included them in the supplement.
Once again, we would like to express our gratitude for the time spent reviewing our manuscript and all the insightful remarks. We do hope that the additions made in the revised version of the manuscript, improved the quality of our paper and that our responses clarified all of the questions raised.
Round 2
Reviewer 1 Report
Comments and Suggestions for Authors
The Authors have addressed all my questions and concerns
Reviewer 2 Report
Comments and Suggestions for Authors
I consider the revised version of this paper suitable for publication